# A qualitative exploration of the educational needs of people living with heart failure: BANDAIDD-Explore study

Caleb Ferguson[1]*, Scott William[1], Sabine Allida[1], Peter S. Macdonald[2], Gary Kilov[3], Clara K. Chow[4], Anthony Keech[5], On behalf of the BANDAIDD Study Investigator team[¶]

1 Centre for Chronic & Complex Care Research, Blacktown Hospital, Western Sydney Local Health District, & School of Nursing, University of Wollongong, Wollongong, Australia, 2 Heart Lung Service, St Vincent's Hospital Sydney & University of New South Wales, Kensington, Australia, 3 Launceston Diabetes Clinic, Baker Heart & Diabetes Institute, University of Sydney & University of Melbourne, Melbourne, Australia, 4 Westmead Applied Research Centre, University of Sydney & Westmead Hospital, Westmead, Sydney, Australia, 5 NHMRC Clinical Trials Centre, University of Sydney & Royal Prince Alfred Hospital, Sydney Local Health District, Sydney, Australia

¶ Membership of the BANDAIDD Study Investigator team is provided in the Acknowledgments.
* calebf@uow.edu.au

**Data Availability Statement:** The data are not publicly available due to containing information that could compromise the privacy of research participants. Due to the sensitive nature of the data

## Abstract

This is an exploratory qualitative study of cardiovascular clinicians, people living with heart failure (HF), and their caregivers. The aim was to understand the unmet educational needs in self-management for adults living with HF. Four focus groups were conducted face-to-face or via videoconference, recorded, and transcribed verbatim for thematic analysis. A total of 23 participants were recruited and included in analyses (clinicians n = 13; adults with HF n = 7; caregivers n = 3). The focus groups were on average 81 (range 73 to 91) minutes in duration. Seven key themes were identified which were: (i) Understanding and reinforcing the signs and symptoms, self-management, medications, and prognosis and severity of HF; (ii) Providing concise and timely education; (iii) Building trust and relationships; (iv) Accessibility of education to support patient needs; (v) Engaging family members and informal caregivers; (vi) Tailoring education to patients diverse needs; and (vii) Navigating the health system and dealing with continuity of care. There were several unmet educational needs for people living with heart failure and their caregivers. Providing patient-centred education is critical to developing understanding and reinforcing the signs and symptoms, prognosis, and severity of heart failure, to underpin self-management and optimise medication adherence. Clinicians, patients, and their caregivers provided several suggestions for improvement, such as the importance of providing concise and timely education and building trust and relationships between clinicians and patients. Priorities for education improvement were also provided, including regional and rural considerations; engaging informal family caregivers; tailoring to culturally and linguistically diverse and indigenous peoples, navigating the health system and ensuring continuity of care.

collected for this study, and in keeping with Sydney Local Health District's (SLHD) Royal Prince Alfred (RPA) Human Research Ethics Committee (HREC) approval, requests to access the dataset from qualified researchers trained in human subject confidentiality protocols may be sent to the corresponding author (calebf@uow.edu.au) and the approving HREC (SLHD-RPAEthics@health. nsw.gov.au).

**Funding:** Prof Caleb Ferguson is supported by a National Health and Medical Research Council 2020 Investigator (Emerging Leadership) Grant 2020–2025 Ref: 1196262. Australian government's Medical Research Future Fund. Prof Anthony Keech is supported by National Health and Medical Research Council Senior Principal Research Fellowships (APP 1137071, & APP 2018537) and an NHMRC Program Grant (APP 1105467). Clara K. Chow is supported by a National Health and Medical Research Leadership Investigator Grant (APP 11195326). This study was funded by the Medical Research Future Fund (MRFF) Cardiovascular Mission Grant Ref: 2009251; Digital solutions for heart failure best practice care. The funders had no role in study design, data collection and analysis, decision to publish, or preparation of the manuscript.

**Competing interests:** The authors have declared that no competing interests exist.

## Introduction

Heart failure (HF) is a common, progressive, and debilitating syndrome, affecting approximately 64 million people worldwide [1]. It is currently the leading cause of hospitalisations and deaths in developed countries. In Australia, HF contributes 1.5–2% of the national health expenditure, with hospital admissions accounting for two-thirds of these costs [2]. Frequent hospital readmissions and emergency department visits are the most preventable cost component [2, 3]. The common causes of rehospitalisation include medication and dietary non-adherence, delays in symptom recognition and response, and a lack of knowledge and skills for self-care. These are known contributing factors to clinical decompensation [4, 5]. The main priorities of care for people with HF involve early recognition and treatment of symptoms as delayed intervention greatly contributes to reduced quality of life, increased morbidity, and mortality [6].

Self-care plays a critical role in promoting success in the treatment of HF [7]. Recognition of signs and symptoms is a fundamental aspect of self-care [8]. Other components include evaluating change, deciding to act or seek help, implementing a treatment strategy, and then evaluating the treatment implemented [8]. To adjust their lifestyle and to care for themselves effectively, people with HF need to be equipped with adequate knowledge and skills [9]. Before people with HF and their informal caregivers are ready to actively participate in decision-making processes, they must be comprehensively educated on both their condition, the available therapeutic options for management and the risks and benefits of each treatment option [10]. Supporting patients to enhance their self-care behaviour through education and learning can lead to a positive effect on lifestyle modification, on response to worsening symptoms and on coping with a chronic syndrome [9]. To enhance the effectiveness, education should be tailored for each patient and family [11]. Support and education are important components of HF care, and must be maintained for as long as necessary in the home setting to address a safe and effective transition from hospital to home [12, 13].

**The BANDAIDD study** (*B*eta-blockers; *A*CE inhibitor or ARB; *N*itrate-hydralazine/ Neprilysin inhibitors; *D*iuretics; *A*ldosterone Antagonist; *I*vabradine; *D*evices (automatic implantable cardioverter defibrillator, cardiac resynchronisation therapy or both; *D*igoxin), is a clinical trial designed to inform patient education to improve self-care using personalised 'e-TIPS' delivered via text messages weekly. This study is registered on the Australian New Zealand Clinical Trials Registry (ID: ACTRN12623000644662) [14].

This sub-study of BANDAIDD, known as BANDAIDD-Explore aimed to explore the educational needs of people living with HF from the perspectives of cardiovascular clinicians (nurses and doctors), adults with HF, and their family caregivers.

## Methods

### Design

This exploratory qualitative study was conducted through a series of two focus groups undertaken with cardiovascular clinicians, adults living with HF and their caregivers. The qualitative method of thematic analysis [15] was used to identify, analyse, organise, and report themes found within the focus groups to elicit educational perspectives of people living with HF. This study reporting of this study was guided by the consolidated criteria for reporting qualitative research (COREQ) S2 Appendix COREQ Checklist [16].

## Sites and participants

Purposive sampling was used to recruit cardiovascular clinicians, adults living with HF, and their caregivers. Multidisciplinary clinicians working primarily in the cardiovascular specialty were sent an invitation to participate through list-service email distribution at participating hospitals (see **S1 Appendix**) across metropolitan Sydney and regional New South Wales, Australia. An invitation to participate was also distributed through email via the Australian Cardiovascular Nursing College (ACNC) and also the Cardiovascular Nursing Council, and Allied Health Council of the Cardiac Society of Australia and New Zealand. Community-dwelling adults living with HF and their caregivers were invited to participate, through study posters placed in the HF outpatient clinics of participating hospitals. Inclusion criteria were as follows: adults with a primary diagnosis of HF with reduced ejection fraction (HFrEF), as per the National Heart Foundation [2] and European Society of Cardiology guidelines (patients) [3]; the ability to participate in face-to-face focus groups or via videoconference; and the ability to communicate and consent in English.

## Focus groups

Four qualitative focus groups were conducted by two experienced qualitative researchers with backgrounds in nursing and medical science (CF and AK) one group was conducted face-to-face in a confidential meeting room and three groups were conducted by videoconference, between August and October 2022. The focus groups were audio recorded and transcribed verbatim by a professional external transcription service. Semi-structured focus group guides were informed by existing literature and developed from the research teams' own experience of working with the patient group and were peer reviewed. Refer to patient carer, and clinician focus group guides outlined in **Tables 1 and 2** below.

## Data analysis

Data collection continued until no new concepts were found, and saturation had occurred. Thematic analysis was undertaken between March and June 2023 according to the methods proposed by Braun and Clark [15]. Two researchers independently reviewed transcripts, coded for meaning units to generate categories and themes. Disagreements were resolved through discussion and consensus (SW, SA & CF). NVivo v12 (released in March 2020) was used to organise and code data. Themes and categories are summarised as results.

**Approvals and ethical considerations.** Each participant received written and verbal information about the research project and provided informed written consent before the

**Table 1. Patient focus group guide.**

| INTERVIEW GUIDE |
| --- |
| 1. What is your experience of receiving patient education about heart failure?<br> a. How and when has this occurred? |
| 2. What topics have been addressed in previous education about your heart failure? |
| 3. What do you think is important information to know about your heart failure and its treatment and management? |
| 4. What is important for you to know about your heart failure? |
| 5. How and when would you like to receive patient education?<br> a. By whom and how often?<br> b. How would you like to be educated? |
| 6. We are designing a new education program for heart failure, what do you think the priority topics for education should be? |
| 7. Is there anything that is important to address about heart failure, that is currently not well addressed in patient education? |

**Table 2. Clinician focus group guide.**

| INTERVIEW GUIDE |
| --- |
| 1. What is your experience of educating patients about their heart failure? |
| 2. How are patients currently educated about their heart failure? |
| 3. What do you feel are the core components of heart failure patient education? |
| 4. What are the priority content areas that need to be addressed? |
| 5. What are the most important topic areas to address? |
| 6. How and when do you think patients like to receive education? |
| 7. Do you feel there are any missing or unmet needs with current patient education? |

focus groups commenced. The research project was approved by Sydney Local Health District (Royal Prince Alfred Hospital) HREC (Approval number: X21-0484). The participants were informed about confidentiality and anonymity of data, and ability to withdraw.

## Results

### Participant characteristics

A total of 23 individuals participated across 4 face-to-face or virtual interviews including 13 clinicians (10 female/3 male), 10 adults with HF and their caregivers (3 female/7 male). The clinician cohort comprised of cardiologists and nurses with varying qualifications and expertise. The cardiologists were consultants engaged in provision of HF inpatient care and/or outpatient clinic-based care. The nurses were clinical nurse specialists or nurse practitioners. Individuals with primary diagnosis of HFrEF were recruited from the outpatient HF clinic setting. Focus groups were on average 81 minutes in duration (range 73 to 91 minutes).

The first 2 focus groups comprised of cardiovascular clinicians (4 cardiologists and 9 nurses). The subsequent 2 focus groups included people living with HF and their carers (7 adults with HF and 3 caregivers).

### Themes

Seven key themes were identified: (i) Understanding and reinforcing the signs and symptoms, self-management, medications, and prognosis and severity of HF; (ii) Providing concise and timely education; (iii) Building trust and relationships; (iv) Accessibility of education to support patient needs; (v) Engaging family members and informal caregivers; (vi) Tailoring education to patients diverse needs; and (vii) Navigating the health system and dealing with continuity of care. A graphical representation of the unmet education needs for heart failure self-management can be seen in **Fig 1**.

**(i) Understanding and reinforcing the signs and symptoms, self-management, medications, and prognosis and severity of HF.** Signs and symptoms, self-management, medications, and prognosis and severity of HF have been considered together as they impact on functional status and quality of life of people with HF. Clinicians highlighted the need for reinforced education around monitoring for signs and symptoms such as leg or ankle oedema, dyspnoea and weight gain that may signal deterioration.

*because we can give them the education about the management side of things, but ultimately, what we really want to do is for them to be able to recognise their signs and symptoms, and to be able to pick up the phone and call us. That's the big thing. We can be monitoring weekly, fortnightly, monthly. Then, sometimes you might not see them for a month, six weeks, but you really want them to be able to recognise whether they're deteriorating, and for them to call you, and to remember to call you.–Nurse, FG2.*

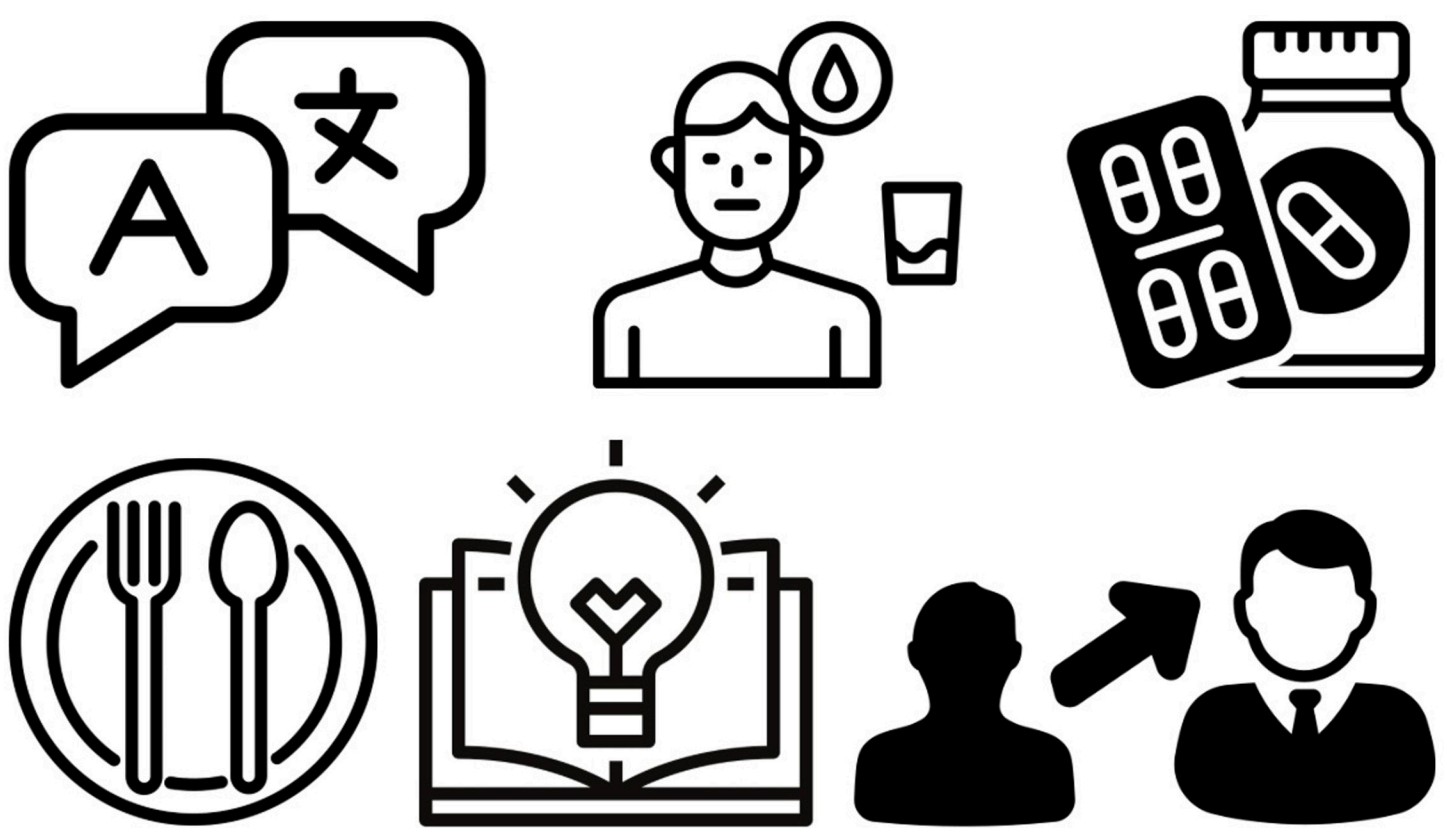

**Fig 1. Graphical presentation of the primary findings in unmet needs in HF self-management education.**

There was also an emphasis on ensuring patients escalate or communicate these symptoms to clinicians as early identification of deterioration may improve outcomes and lessen the intervention required to stabilise patients.

*Do you think most of them understand that they need to ring the alarm bells if they've put on two or three kilograms?–Cardiologist, FG1.*

*When I do my education, when I meet my patients, my priority is basically, there's to do, and to watch out for. I have a to watch out for list, and then what to do about it list. To watch out for is basically weight gain, solid legs, or oedema, and breathing problems. I say it in those lay terms. Then, to do list is, must do every day, is fluid restriction, 1.2, 1.5, one litre, whatever they're on. Taking their medications every day and weighing themselves every day.–Nurse, FG2.*

Patients and clinicians agreed that medications are a fundamental component of HF education as patients with HF are prescribed a lot of medications, which are also frequently complex. Educating patients on how the medications work, the rationale for them and common side-effects and interactions with other medications, together with the significance of signs and symptoms may improve understanding and improve medication adherence. Patients mentioned using the internet, i.e., Google, to find out general information and check medication side effects to inform or drive discussions with their caring clinician.

*I suppose to make sure they know what they're taking and what the pills do to try and keep them adherent to the regime, taking them at a regular time, the right doses. The physical boxes and bottles, making sure [they're not using] old ones and that they're important to continue.–Nurse, FG1.*

*Probably the most difficult thing is the amount of medications. I like to research all that, so for me, they would leave my new medications and trialling different things, and I'd go on Google and start Googling them all to find out more about the product and have my questions ready for next time.–Male Patient, FG3.*

Clinicians reported that repeating important information around self-care and communicating signs of deterioration reinforced learning for HF patients. The timing of education is also important when considering their capacity to take in and retain information.

*I think short and simple most of the time is the best way for a lot of patients because you see them glaze over, so keep it - not short, sharp but quite simple. Don't bombard them with too much information because they just dread it quite often they're not interested. . . . You might give them a lot of education but I find most of them, keep it simple. Simple's the best [in moderation]. Just the basics of what they need to know about the heart, how it works but very simply and also what to expect from the [like the] daily food restriction and why the most important thing is take the actions and escalating. Escalate, escalate, escalate, that's the whole idea of us.–Nurse, FG1.*

*They're the themes that I try and use to break down the information. Again, very big, obviously, on trying to reinforce that with repeated communication.–Cardiologist, FG2.*

Another key focus for clinicians was stressing the seriousness and severity of heart failure to emphasise the importance of self-management and medication adherence. Patients value reinforcement in communicating the eventual deterioration, which is part of an HF diagnosis, and the importance of this for self-management and care.

*The other thing I tell my patients is this is a really serious condition. If you don't take it seriously, you're either going to end up in hospital or die. The medications can help you feel better but it's really important you take them. If you explain how important a serious heart condition is, they're more likely to go on the journey with you. The last thing I say on the first visit when they come to the clinic to see us is, I look forward to helping you. That really makes a big difference to the patients because they're often on their second, third or fourth cardiologist and they've become a bit disillusioned with the system. To be able to try and get confidence into them and explain that this is a big deal; this is not just going to go away and get better. This is something that's a lifetime-defining illness. I say that to them up front and they're like, I didn't realise it was that bad.–Cardiologist, FG1.*

*They're going in for a chest infection and then you get better or having your gallbladder out and then you're better. You have to be quite blunt sometimes, very blunt. If you don't stop drinking alcohol, you're going to die.–Nurse, FG1.*

Due to the overwhelming nature of being diagnosed with a life-limiting syndrome, good bedside manner was thought to be key to providing quality support and reassurance.

*So on top of everything else that was going on, it was just, one, understanding and making sense of the reality of what it meant and what the journey forward looked like, which was also*

*further challenged by the fact that I was getting such a variety of spectre of the possibilities from one doctor telling me that my life as I knew it was over and that I'd never get out of bed again, all the way through to [Clinician], who kept reminding me that everyone has their own journey.–Female Patient, FG3.*

*The way she says, yes I agree you're losing weight. Encouraging me to weigh myself every day. I've got really bad nerve damage in my legs, which happened when I had the stroke and I had cellulitis major infection while I was in hospital. The legs have never recovered.–Patient, FG4.*

There are difficult and important aspects in educating patients and carers about options for end-of-life care. The trajectory of HF is not necessarily predictable.

*Another thing that I wrote down about gaps, was palliative care. usually, within the second or third appointment that I have with the patients, I try and talk about wills, enduring guardianship, power of attorney, introduce advance care directives, and will help them through that. I often find, when they get to that point, where I think palliative care would be useful, sometimes it's not always what you think it will be for the patients and I still don't know if heart failure's completely understood that well, and general palliative care may be a little bit different. I find that palliative care is slightly more difficult for heart failure than it is, probably, for cancer, and it's because of that trajectory, where their progress goes up and down, and there's no real end date.–Nurse, FG2.*

*Yeah, I got really scared - he was told end of life care, once, and everybody thought he was dying. But one medication saved him, and was just a really big gamble. At the moment, he was actually very weak, but he just won. He just won that gamble, and he just survived. Yeah, everything going back to normal.–Male Carer, FG3.*

**(ii) Providing concise and timely education.** Patients commonly reported feeling overwhelmed described as receiving a *'tsunami of information'* or being on *'a roller coaster ride'*. They reported that the more education they received, sometimes the more confused they became. The overall experience felt like a bombardment of information during a time of uncertainty. Uncertainty related to the implications this diagnosis could have on their lifestyle and the prospect of living with this chronic condition. There is a need to consider the readiness to learn from a patient's perspective.

*Because I'd never even understood that heart failure was a thing, because even if you go to the Heart Foundation website now, it's all heart attack. It's impossible to find stuff on heart failure there. I look back at my notes now and I can see, very clearly, with the benefit of hindsight, what it was that [Clinicians] were telling me, and why it mattered, but at the time I didn't know what I didn't know, and I didn't know what was important.–Female Patient, FG3.*

*it was just very overwhelming to hear all these different things about the heart failure, but I was remaining as positive as I could. From an education perspective, I think initially when you find out, you have all these doctors and nurses talking to you, and it's very overwhelming. I was probably in denial a little bit, too. it was a rollercoaster ride, but I would have to say that the support and the education part of it was amazing.–Male Patient, FG3.*

*Maybe getting some more feedback from them. Sometimes I feel like I'll throw a lot of information about fluid restriction and weight to patients and I'm not really sure how much they're taking in.–Nurse, FG1.*

**(iii) Building trust and relationships.** Clinicians reported the difficulty of providing initial education and building rapport with patients pre-discharge due to limitations in the clinical environment. Patients agreed that home visits are a valuable approach to providing relevant and individualised education, facilitating regular communication, and building trust.

*My managers, are trying to get me to put discharge dates against each patient and have a turnover of one–of three months. Which is quite often not realistic for a lot of the patients that I see, particularly the ones that are really unstable or fragile heart failure patients. Continuity of care is vital . . . The important thing is to go on to establishing a long-term relationship with them. It's you that you want them to call, not the ambulance, if things start to go wrong at the beginning.–Nurse, FG2.*

*I think that the key for me was someone coming into my own home and making me feel more comfortable. [Nurse] - she was very honest and went that extra mile in relation to putting systems in place for me that were tailored for my needs, to make sure that I had the best recovery possible. I couldn't have done it without that. Even calling into my shop and taking my blood pressure at different times in the day to go, okay, your shop has this impact on you, when you're doing these tasks, it was just so helpful. Where I suppose old-fashioned nursing, none of this would have happened, but that relationship was just so powerful and meaningful to me that I didn't feel alone.–Male Patient, FG3.*

Clinicians and patients share a mutual responsibility when establishing rapport to facilitate effective escalations of signs and symptoms.

*We don't have resources for us to be on the road, and then do inpatient, but also, retainment during an acute admission is very minimal, and you're just going to repeat yourself. So, home visit within those first couple of days, post discharge, will give them the start of the education rolling. If it's somebody that you're going to keep going to, then it's because they can't self-manage, or they've got a high possibility of readmission for an exacerbation, and you're working on those keystone things that will get them to ring you, or increase compliance.–Nurse, FG2.*

When building relationships with patients, it was important to note that routine and consistency are vital in the continuity of care and in preventing deterioration.

*according to the data, when one of our nurses goes on holidays, readmission rate goes up and when they come back, the readmission rate goes down.–Cardiologist, FG1.*

*We've been lucky enough to have continuity of care, the same cardiology and the same nurse, and if that nurse goes away, she makes sure we know who is relieving her on that week. If we are in dire straits, we can organise for a home visit, and we have in earlier days, and having the same person who knows us within the home is pretty important, and knows what our domestic situation is. I call that continuity, both in the home and in the clinical scene.–Female Patient, FG3.*

**(iv) Accessibility of education to support patient needs.** Education should be tailored to the needs of patients who may be isolated due to mobility difficulties in a way that ensures equal access to self-management information.

*For me, family are essential, doing that, just because too, even getting them to appointments, and things like that. For a heart failure patient, that's quite difficult for them. Some doctors' offices might have 10 steps up. They can't get there. So, it's really good to involve family. I suppose I'm lucky enough that I do work in an area where there's a lot of family that have support, but when they don't have support –. . ...I ask them, do you have any support? What's your supports at home? If they don't, then try to arrange that for them, so they can get transport to an appointment, and things.–Nurse, FG2.*

*If they're socially isolated, I would definitely do more frequent visits than someone who has more social support and family support too. Because they are probably less likely to call if there's a problem. Whereas someone who has family; daughters, sons popping in all the time or living with them, would make the call, rather than the patient. That's my experience.–Nurse, FG1.*

*I've got mobility issues, so I use a walker to get out and about. The last few weeks here in the bush, we've had a lot of rain. It's very hard for me to get the walker into a taxi, then out. You end up saturated.–Male Patient, FG4.*

The benefits of regular home visits allow for detailed and relevant observations of patients', environment, signs and symptoms and reinforcement of actions and escalations.

*The minute you open the door you go, oh dear. That's because you always look at their ankles straight away, their legs and you assess their breathing the minute they open the door. If you're doing telehealth, you don't get the same approach because that's it, they're waiting for your call, they're really eager. Often they don't have Zoom so you probably can't do face-to-face by phone with them because they don't have the right equipment.–Nurse, FG1.*

*Which is the beauty of going into the home. It's the same with their medications. I think you can actually look visibly at their medications, look at Webster-paks, Dosette boxes, packets. We've got patients that you walk in and there's medications scattered all over the table. So, you've got an idea then of what's going on at home. I think then, from that, getting that real perspective of the home life, that gives you a good sense of where the education needs to be targeted.–Nurse, FG2.*

Individualisation of care to maintain continuity of care could be adapting and adjusting for experiencing homelessness or lacking social support.

*One of my patients that I only see in the foyer, he was homeless but he would come into the foyer so we would see him in the foyer. He didn't like anything that was going to be an actual clinical scenario but he could handle the foyer at the old buildings church there and chat.–Nurse, FG1.*

*If they're socially isolated, I would definitely do more frequent visits than someone who has more social support and family support too. Because they are probably less likely to call if there's a problem. Whereas someone who has family; daughters, sons popping in all the time or living with them, would make the call, rather than the patient. One of the other things I'll often call is if my patient's had some form of social disaster, something's gone wrong in your life.–Nurse, FG1.*

There are numerous unmet self-management needs for HF patients who live in regional or rural areas. These include social isolation and the difficulty of maintaining continuity of care.

*You take what the doctor gives you. I've had good health service and I've had some very bad service. you rely on what the doctor - for your condition - says you're going to take X. You've got to remember, we're out here in the bush. To get into your GP, you could be three weeks out. Your specialist, you're lucky to see him once every 12 months because there's just not enough quality specialists out in the country. We're a big centre now. ... The hospital's probably looking after 200,000 people. We're not just a town of 40. But the only other information I've got on medications is Dr Google.–Male Patient, FG4.*

**(v) Engaging family members and informal caregivers.** The educational needs of caregivers should be considered in parallel to patients as engaging caregivers and family members may contribute to improved outcomes and knowledge. Both patients and clinicians agreed that family members and caregivers play a critical role in the patient's HF care.

*Having different resources in different languages. Also maybe involving some of the family members or carers who they trust and have confidence with to communicate things that I might not be able to communicate.–Nurse, FG1.*

Informal caregivers and family members provide direct care to the patient and are perfectly placed to influence and support self-management and lifestyle behaviour change.

*I can't remember, I was too busy, I was too sick, everything was happening, there were people sticking needles into me. That visit at home really makes a difference, they're in their environment, they're feeling better. They've often got their carer with them. Involving the carer is super important because so many times the wife at the hospital will say, did you take your tablets this morning? So they're able to monitor compliance and assistance with medications and to remind them of things.–Cardiologist, FG1.*

*Maybe I'm lucky in the sense that I've got my wife, who is assisting me in that sense. Therefore it's not as hard if you're on your own. That's probably a big help, having someone that you can share that journey.–Male Patient, FG4.*

**(vi) Tailoring education to patient's diverse needs.** There was an emphasis on providing resources that are culturally and linguistically tailored as it can be challenging to provide adequate HF education to patients from diverse backgrounds. It can be useful to provide education to the patients' carers who may have a greater comprehension of the English language.

*You know the biggest challenge I found is communicating with non-English speaking background patients, which actually makes up quite a lot of our list. When you go to their house and try not to inundate them with too much information and it's really helpful that we have action plans in different languages. Even if that's the only thing that I can get across, it's the signs and the symptoms to look out for and to contact and escalate to your doctor or nurse as soon as possible.–Nurse, FG1.*

*Making sure that I get all the information across that I'd like to, but again, trying to tailor that to the individual and their individual diagnosis, their sociocultural situation, all those factors obviously need to be taken into account.–Cardiologist, FG2.*

*From the [Hospital], there is a CALD population, lower socioeconomic, it's sometimes quite hard to educate them on the first go. So, I like to reinforce things on my second, third, fourth,*

*fifth visit. I really welcome families, because it's a population that has a lot of family involved in the actual patient's care. I find that a lot of our patients are older.–Nurse, FG1.*

Understanding food, dietary, and medication interactions which could be tailored for diverse patients.

*I'm saying that the information that I received, for example, here's what you can eat with Warfarin, was all based on a very English diet, and every time I tried something new, like ginger or - green tea is not a good example, but things that would never have expected to have thrown my Warfarin levels out, because they weren't on the list, we'd then discover that in fact, well, that's actually not a really good thing to eat. applying a wider filter, given the diversity of diet that we have in Australia.–Female Patient, FG3.*

Resources do currently exist which are offered in multiple languages from the Heart Foundation.

*I think they do have a relatively comprehensive list of different languages that they can convert the Heart Foundation action plan into, which is really helpful for patients. there's a big booklet that goes through things pretty comprehensively, I think it does strike a good balance of providing information, but not too much.–Nurse, FG2.*

There are concerns about maintaining medication adherence for patients due to financial difficulties. Furthermore, there are multifaceted barriers to accessing government assistance.

*Well, I stopped the lot, because they were just so expensive, and I was paying $300 a month at least for medicines, and while I was working, I could afford that. But once I retired, I was on a shoestring, so I had to decide what to do, and sadly, I decided to cut it out. But in the end, good things came out of bad, where I collapsed but then came under care, and now with the help of pension rates and so on, I'm on lesser medicines, but better medicines.–Male Patient, FG3.*

*I find it more challenging with different multicultural groups. As you said before, hugely diverse on a multicultural front, but also on a socioeconomic front, a lot of the challenges we face is patients run out of money to continue their medications. So, part of our service is about having a social worker involved in the clinic, that can help support them with navigating Centrelink, really just to help with a whole bunch of issues, but one of those issues is helping them to stick with medical therapy, to afford it.–Nurse, FG2.*

*There's a lot of money spent each month on medication on our part because we're not on any benefits. The pharmacy does say, you can apply for this or you can apply for that. There's no information at all on what you can do to get some assistance. Because it's long-term medication. It's a long-term expense.–Female Carer, FG4.*

The commitment and considerations for practices in maintaining continuity of care with patients who identify as having an Indigenous background are of high importance. Important considerations include building trust and rapport with patients from Indigenous backgrounds, which includes providing the services of an Aboriginal healthcare worker.

*I just want to also mention about the Indigenous patients, and the role that offering an Aboriginal healthcare worker to go with you on a visit is very worthwhile, and a great way to*

*start a rapport that might last quite a long time, because engagement with the Indigenous population is always quite difficult. I find if we can have that, follow through. So, if you say, let's do something, you've got to do it, and you've got to show that you are doing it. Then you will get a great rapport with this group.–Nurse, FG2.*

*I do an Aboriginal outreach clinic. It took me about two years, to gain the trust. you just can't turn up. You need to have a sustained connection with them, with the community. I've been going out there for 16 years now, and I think now, they do come and see me. When I first started going there, half the waiting list would be disappearing when they knew I was coming to town, but now they all tend to come.–Cardiologist, FG2.*

**(vii) Navigating the health system and dealing with continuity of care.** The experiences of patients differ depending on the healthcare setting and the clinician providing the treatment. The resources for HF education can vary by setting. Patient and informal carer's understanding of the healthcare system is important when maintaining continuity of care.

*But that was a bit of a shock, though, if you don't mind me interrupting, going from one cardiologist who didn't say, ah, you're really not looking too good, to another cardiologist who says, well, I think you need a pacemaker and I think you need defibrillator. I think you need them now, but if you go through the public health system, well, you'll be waiting a long time. There was no beds, but we're privately insured, so within about three days, [patient] was in hospital and had a pacemaker. . . –Female Carer, FG3.*

. . .and the defibrillator. So that was a bit of a shock. There was no preparation between medication and what I consider conservative treatment to suddenly, defibrillator, pacemaker, ASAP. That was - there was a gap there, I thought, an information gap, so to speak. Yeah, I think that was somehow. . . –Female Carer, FG3.

*It's difficult, because one of the gaps I was actually going to raise was nursing home patients, because we find, unfortunately, that's an unmet need there, repeated admissions. So, we say, oh, they're being looked after by someone, that'll be okay, but in reality, that's not how it happens. Some of the things we've been trying to do is upskill the nursing homes, just in terms of a phone call to the nurses, the RNs looking after the patients, just to give them a heads-up about the discharge, what's in the discharge summary. Also, cluing them in that there's a flexible diuretic regime, generally, included in the discharge summary, so they've got access to that. They don't need to wait for the GP to phone. Just empowering them to use things that are available.–Nurse, FG2.*

## Discussion

The key findings from this study highlight the need to prioritise patient-centred approaches to education, addressing patient unmet needs and enabling patients and caregivers to enhance self-care. Common to all participants was the struggle with awareness of HF and initial understanding of the importance of monitoring signs and symptoms of deterioration, adherence to medication and self-management. This is concerning as optimal management of HF relies heavily on how effectively patients are able to self-care. A recent editorial by Ivynian and colleagues suggests that this is partly due to the confusion and misconceptions around the meaning of the term 'heart failure' [17]. The use of this term can create fear and anxiety [18] similar to the term 'cancer', which has historically been avoided in patient-provider communications.

The fear and anxiety associated with this term can contribute to denial or non-acceptance of their condition and as a result, interfere with patient's ability to perform crucial self-care practices [19, 20]. Ineffective self-care often leads to worsening of HF and accounts for approximately half of preventable HF-related hospitalisations [21].

The term 'heart failure' can also cause confusion around its chronic nature and associated symptoms. Patients may not easily recognise the difference between HF and other cardiac conditions such as a heart attack or that symptoms not directly 'felt' by the heart are a signal of deteriorating heart function [22]. As a result, symptoms such as breathlessness, fatigue or weight gain may be perceived as less threatening compared to chest pain associated with a heart attack [23]. This has major implications for patient outcomes as delays in symptom recognition and care seeking can lead to more complex treatment and longer lengths of hospital stay. The association of HF with heart attack also contributes to misconceptions about HF being an acute problem and an illness that could be overcome [24]. Consequently, ongoing self-management may not be perceived as important or relevant. This emphasises the importance of ensuring that patients not only understand what HF is and isn't, its seriousness, chronicity, and management when delivering education but also reinforcing this consistent information across the HF care continuum.

Findings from this study are consistent with previous evidence that demonstrates the effect of previous healthcare experiences on care-seeking and symptom management in HF and underscores the significance of building trust and relationships with providers to ensure continuity of care [25–28]. Maintaining continuity of care is very important to patients as this provides a sense of safety knowing that they are being treated by long-term providers with whom they had established a trusting relationship. Given their previous involvement in patient care, long-term providers are perceived to be better equipped to manage the patient's condition effectively and also minimise the risk of negative consequences associated with being managed by multiple providers. This is supported by another study which showed that a positive patient-provider relationship and continuity of care facilitate care seeking as it provides a sense of confidence and support [29]. Trust in clinicians facilitates access to healthcare and disclosure of relevant information and thereby supports accurate and timely diagnosis to be made. Whereas negative patient-provider relationships affected the capacity and motivation of patients to self-manage [29].

There are limited evidence-based mHealth educational interventions with good effect in relation to heart failure knowledge, self-efficacy, self-care, health-related quality of life or heart failure-related hospitalisation [12]. Further, ensuring mHealth interventions deliver content which adequately addresses the gaps in self-management is important [30]. Robust clinical trials of theory-driven, self-care digital interventions that assess short—and long-term clinical outcomes, such as rehospitalisation, mortality, and patient-reported outcomes, are needed [31]. Adapting traditional paper-based information and hosting it on a website or in a static application likely has a very limited effect. Future digital health interventions must be underpinned by health behaviour change theory and consider functionality, such as nudged, reminders, and interactive gamification [31]. For example, electronic health interventions in adults living with heart failure have shown improvements in self-care with mobile phone accessibility noted as an integral component of usability [32]. Ensuring mHealth interventions are designed for use by diverse user groups is an essential component of research and development [33].

Lastly, our results also emphasise the critical importance of engaging family or informal caregivers as an approach to optimising self-care behaviour and outcomes. This is consistent with existing work that highlights the success of providing family-centred interventions in the context of heart failure to enhance self-care behaviour and reduce hospitalisation [34, 35].

However, few didactic studies in heart failure have addressed digital interventions [36]. Future digital approaches to heart failure education should consider targeting caregivers as well as individuals living with heart failure. The national and international implications of this work include understanding the complex and diverse needs of adults living with heart failure, relevant to their own experiences, cultural and personal preferences, and health literacy relevant to navigating their local health systems. This may be used to guide future work to further develop policy and practice to ensure greater accessibility of heart failure self-management education.

## Limitations

This study included primarily expert clinicians, heart failure cardiologists and specialist nurses and patients and caregivers from both metropolitan and regional settings in New South Wales, Australia. The findings of this study may have limited transferability. Further, as all focus groups with patients and caregivers were conducted online using video conference, it is likely that this sample has high levels of digital literacy, which may have influenced some findings. Whilst ethnicity and diversity data were not collected, most participants were of Caucasian background, which may limit the richness and diversity of perspectives.

## Conclusion

This study revealed that there were several unmet educational needs for people living with heart failure and their caregivers. Providing patient-centred education was thought to be critical to developing understanding and reinforcing the signs and symptoms, prognosis, and severity of heart failure, to underpin self-management and optimise medication adherence. Clinicians, patients, and their caregivers provided several suggestions for improvement, such as the importance of providing concise and timely education and building trust and relationships between clinicians and patients. The clinical implications of this work emphasise the importance of prioritising self-management. This includes ensuring accessibility of education to meet the diverse needs of heart failure patients, navigating the health system, and ensuring continuity of care.

## Supporting information

**S1 Appendix. BANDAIDD study team investigators.**
(DOCX)

**S2 Appendix. COREQ (COnsolidated criteria for REporting Qualitative research) checklist.**
(PDF)

## Acknowledgments

Thanks to Kaitlyn Griffin, Portia Westall and Nicola Barrie for support with focus group organisation and administration, and to the many nurses and patients and their caregivers for their insights, feedback and time. Consumers L-JL and GB are members of the BANDAIDD Research team who received the MRFF support listed in the **S1 Appendix**.

BANDAIDD Study Investigator team - Anthony Keech, Sean Lal, Peter Macdonald, Caleb Ferguson, Christopher Ryan, Alicia Jenkins, Kathleen Dempsey, Clara Chow, Rachel O'Connell, Gary Kilov, Rebecca Mister, Sandy Middleton, Douglas Drak, Jo-Dee Lattimore, Andrzej Januszewski. Study team lead: Professor Anthony Keech, anthony.keech@sydney.edu.au

## Author Contributions

**Conceptualization:** Caleb Ferguson, Peter S. Macdonald, Gary Kilov, Clara K. Chow, Anthony Keech.

**Data curation:** Caleb Ferguson.

**Formal analysis:** Scott William, Sabine Allida.

**Funding acquisition:** Caleb Ferguson, Peter S. Macdonald, Gary Kilov, Clara K. Chow, Anthony Keech.

**Writing – original draft:** Caleb Ferguson, Scott William, Sabine Allida, Anthony Keech.

**Writing – review & editing:** Scott William, Sabine Allida, Gary Kilov.

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
