## [Decision Letter · Decision Letter 0]

10 Sep 2024

PONE-D-24-23011A qualitative exploration of the educational needs of people living with heart failure: BANDAIDD-Explore study.PLOS ONE

Dear Dr. Ferguson,

Thank you for submitting your manuscript to PLOS ONE. After careful consideration, we feel that it has merit but does not fully meet PLOS ONE’s publication criteria as it currently stands. Therefore, we invite you to submit a revised version of the manuscript that addresses the points raised during the review process.

The study is well-written and supports existing evidence regarding educational gaps for heart failure patients. However, some minor revisions should be addressed before proceeding with publication.

We look forward to receiving your revised manuscript.

Kind regards,

Francesca Ferrè

Academic Editor

PLOS ONE

Journal Requirements: When submitting your revision, we need you to address these additional requirements. 1. Please ensure that your manuscript meets PLOS ONE's style requirements, including those for file naming. The PLOS ONE style templates can be found at https://journals.plos.org/plosone/s/file?id=wjVg/PLOSOne_formatting_sample_main_body.pdf and https://journals.plos.org/plosone/s/file?id=ba62/PLOSOne_formatting_sample_title_authors_affiliations.pdf 2. Thank you for stating the following financial disclosure: "Prof Caleb Ferguson is supported by a National Health and Medical Research Council 2020 Investigator (Emerging Leadership) Grant 2020–2025 Ref: 1196262. Australian government’s Medical Research Future Fund. Prof Anthony Keech is supported by National Health and Medical Research Council Senior Principal Research Fellowships (APP 1137071, & APP 2018537) and an NHMRC Program Grant (APP 1105467). Prof Clara Chow is supported by a National Health and Medical Research Leadership Investigator Grant (APP 11195326).  This study was funded by the Medical Research Future Fund (MRFF) Cardiovascular Mission Grant Ref: 2009251; Digital solutions for heart failure best practice care." Please state what role the funders took in the study.  If the funders had no role, please state: ""The funders had no role in study design, data collection and analysis, decision to publish, or preparation of the manuscript."" If this statement is not correct you must amend it as needed. Please include this amended Role of Funder statement in your cover letter; we will change the online submission form on your behalf. 3. We note that your Data Availability Statement is currently as follows: All relevant data are within the manuscript and its Supporting Information files. Please confirm at this time whether or not your submission contains all raw data required to replicate the results of your study. Authors must share the “minimal data set” for their submission. PLOS defines the minimal data set to consist of the data required to replicate all study findings reported in the article, as well as related metadata and methods (https://journals.plos.org/plosone/s/data-availability#loc-minimal-data-set-definition). For example, authors should submit the following data: - The values behind the means, standard deviations and other measures reported;- The values used to build graphs;- The points extracted from images for analysis. Authors do not need to submit their entire data set if only a portion of the data was used in the reported study. If your submission does not contain these data, please either upload them as Supporting Information files or deposit them to a stable, public repository and provide us with the relevant URLs, DOIs, or accession numbers. For a list of recommended repositories, please see https://journals.plos.org/plosone/s/recommended-repositories. If there are ethical or legal restrictions on sharing a de-identified data set, please explain them in detail (e.g., data contain potentially sensitive information, data are owned by a third-party organization, etc.) and who has imposed them (e.g., an ethics committee). Please also provide contact information for a data access committee, ethics committee, or other institutional body to which data requests may be sent. If data are owned by a third party, please indicate how others may request data access. 4. One of the noted authors is a group or consortium BANDAIDD Study Investigator team. In addition to naming the author group, please list the individual authors and affiliations within this group in the acknowledgments section of your manuscript. Please also indicate clearly a lead author for this group along with a contact email address. 5. Please include a copy of Table 1 & 2 which you refer to in your text on page 4. 6. Please include captions for your Supporting Information files at the end of your manuscript, and update any in-text citations to match accordingly. Please see our Supporting Information guidelines for more information: http://journals.plos.org/plosone/s/supporting-information. 7. Please review your reference list to ensure that it is complete and correct. If you have cited papers that have been retracted, please include the rationale for doing so in the manuscript text, or remove these references and replace them with relevant current references. Any changes to the reference list should be mentioned in the rebuttal letter that accompanies your revised manuscript. If you need to cite a retracted article, indicate the article’s retracted status in the References list and also include a citation and full reference for the retraction notice.

Reviewers' comments:

Reviewer's Responses to Questions

**Comments to the Author**

1. Is the manuscript technically sound, and do the data support the conclusions?

Reviewer #1: Yes

Reviewer #2: Partly

2. Has the statistical analysis been performed appropriately and rigorously? 

Reviewer #1: Yes

Reviewer #2: N/A

3. Have the authors made all data underlying the findings in their manuscript fully available?

Reviewer #1: Yes

Reviewer #2: Yes

4. Is the manuscript presented in an intelligible fashion and written in standard English?

Reviewer #1: Yes

Reviewer #2: Yes

5. Review Comments to the Author

Reviewer #1: The study from Ferguson and colleagues presents a qualitative exploration of the educational needs of individuals living with heart failure (HF), their caregivers, and cardiovascular clinicians in New South Wales, Australia. This study identifies key themes regarding educational gaps and proposes improvements for patient-centered education to enhance self-management and overall care.

The study is well-written and interesting with some minor revisions that should be considered by the authors:

Abstract

Line 88: "Priorities identified for heart failure education include: timing of education, tailoring education to support the diverse needs of patients and the importance of continuity of care." The colon after "include" should be checked for consistency with other lists in the manuscript.

Introduction

Line 94: "Heart failure (HF) is a common, progressive, and debilitating syndrome affecting approximately 64 million people worldwide (1)." Consider rephrasing to "Heart failure (HF) is a common, progressive, and debilitating syndrome, affecting approximately 64 million people worldwide (1)."

Line 96: "In Australia, 1.5-2% of the national health expenditure is contributed by HF with hospital admissions accounting for two-thirds of these costs (2)." This could be clearer as "In Australia, HF contributes 1.5-2% of the national health expenditure, with hospital admissions accounting for two-thirds of these costs (2)."

Include explicit research questions or hypotheses

Methods

Line 134: "Purposive sampling was used to recruit cardiovascular clinicians, people living with HF and their caregivers." This might read better as "Purposive sampling was used to recruit cardiovascular clinicians, people living with HF, and their caregivers."

Line 138: "Community-dwelling adults living with HF and their caregivers were invited to participate through study posters placed in the HF outpatient clinics of participating hospitals." Consider rephrasing for clarity.

Provide more detail on the coding process and consensus-building in thematic analysis.

Results

Line 178: "Seven key themes were identified: (i) Understanding and reinforcing the signs and symptoms, self-management, medications, and prognosis and severity of HF; (ii) Providing concise and timely education; (iii) Building trust and relationships (iv) Accessibility of education to support patient needs; (v) Engaging family members and informal caregivers; (vi) Tailoring education to patients diverse needs; and (vi) Navigating the health system and dealing with continuity of care." Correct the repeated "vi" and ensure all themes are consistently listed with punctuation.

Discussion

Conclusion

Line 531: "The clinical implications of this work are to prioritise self-management." Consider rephrasing to "The clinical implications of this work emphasize the prioritization of self-management."

Emphasize the broader implications of the findings for policy in the Australian context and elsewhere.

For the COREQ checklist many items are listed as NA, please check if it is possible to add this information in th text, otherwise state them as limits of the study.

The study can benefit from a central figure highlighting the seven themes emerged in the analysis

State clearly that tables 1 and 2 are in supplementary materials.

Reviewer #2: Great subject in particular knowing that patients in your country can get home visits. Interesting that you collected data from the healthcare providers, patient with HF and their caregiver, so this provides a better understanding of each one experience with this disease. Your manuscript is relevant for anyone delivering care to these patients.

it is recommended to add more details/emphasis about the mHealth as it should be added to the discussion/results; more information and evidence are available in particular with HF patients in the Canadian context.

the methodology lacks information about the types of questions you did ask because you were targeting 2 different types of respondents/participants. in addition, you did not complete all the steps or rather explain ensuring the rigor of your analysis.

The abstract does not have the purpose of the study as you start right away with the methodology; details about the purpose should be added.

6. PLOS authors have the option to publish the peer review history of their article (what does this mean?). If published, this will include your full peer review and any attached files.

Reviewer #1: No

Reviewer #2: No

---

## [Author Response · Author response to Decision Letter 0]

14 Oct 2024

Please find Response to Reviewer document uploaded in the system.

The below section has been revised

DATA AVAILABILITY STATEMENT

The data are not publicly available due to containing information that could compromise the privacy of research participants. Due to the sensitive nature of the data collected for this study, and in keeping with the studies human research ethical committee approval, requests to access the dataset from qualified researchers trained in human subject confidentiality protocols may be sent to the corresponding author and the approving Human Research Ethics Committee by emailing the corresponding author calebf@uow.edu.au and the HREC via SLHD-RPAEthics@health.nsw.gov.au

---

## [Editor Report · Decision Letter 1]

5 Nov 2024

A qualitative exploration of the educational needs of people living with heart failure: BANDAIDD-Explore study.

PONE-D-24-23011R1

Dear Dr. Ferguson,

We’re pleased to inform you that your manuscript has been judged scientifically suitable for publication and will be formally accepted for publication once it meets all outstanding technical requirements.

Kind regards,

Francesca Ferrè

Academic Editor

PLOS ONE
---

## [Editor Report · Acceptance letter]

14 Nov 2024

PONE-D-24-23011R1 

PLOS ONE

Dear Dr. Ferguson, 

I'm pleased to inform you that your manuscript has been deemed suitable for publication in PLOS ONE. Congratulations! Your manuscript is now being handed over to our production team.

Kind regards, 

on behalf of

Dr. Francesca Ferrè 

Academic Editor

PLOS ONE